# AutoPSV: Automated Process-Supervised Verifier

**Jianqiao Lu**[1], **Zhiyang Dou**[1], **Hongru Wang**[2], **Zeyu Cao**[3],
**Jianbo Dai**[4], **Yingjia Wan**[3], **Yunlong Feng**[5], **Zhijiang Guo**[3][†]
[1]The University of Hong Kong    [2]The Chinese University of Hong Kong
[3]University of Cambridge    [4]University of Edinburgh    [5]Independent

jqlu@cs.hku.hk, zg283@cam.ac.uk

## Abstract

In this work, we propose a novel method named **Auto**mated **P**rocess-**S**upervised **V**erifier (AUTOPSV) to enhance the reasoning capabilities of large language models (LLMs) by automatically annotating the reasoning steps. AUTOPSV begins by training a verification model on the correctness of final answers, enabling it to generate automatic process annotations. This verification model assigns a confidence score to each reasoning step, indicating the probability of arriving at the correct final answer from that point onward. We detect relative changes in the verification's confidence scores across reasoning steps to automatically annotate the reasoning process, enabling error detection even in scenarios where ground truth answers are unavailable. This alleviates the need for numerous manual annotations or the high computational costs associated with model-induced annotation approaches. We experimentally validate that the step-level confidence changes learned by the verification model trained on the final answer correctness can effectively identify errors in the reasoning steps. We demonstrate that the verification model, when trained on process annotations generated by AUTOPSV, exhibits improved performance in selecting correct answers from multiple LLM-generated outputs. Notably, we achieve substantial improvements across five datasets in mathematics and commonsense reasoning. The source code of AUTOPSV is available at https://github.com/rookie-joe/AutoPSV.

## 1 Introduction

Large language models (LLMs) have shown impressive performance on various reasoning tasks [1–4]. Prior efforts primarily focus on specific prompting techniques, such as few-shot prompting with intermediate steps and augmented demonstrations [5–8]. While these methods have shown promise, their effectiveness is often task-specific, and designing prompts can be labor-intensive, leading to inconsistent results [9, 10]. Another approach to improve reasoning in LLMs is through instruction tuning or knowledge distillation [11–14]. These methods typically involve fine-tuning LLMs and require a large set of examples annotated with chain-of-thoughts (CoT; [5]). However, these approaches can be resource-intensive and may not always produce reliable results.

To address these challenges, verification techniques have emerged as a promising solution [15, 16]. Verification models are trained to evaluate and potentially correct the reasoning process generated by LLMs. This approach aims to mitigate the risk of relying solely on the top-1 result, which may not always be reliable [17, 18]. By reranking candidate responses, verification models can ensure higher accuracy and consistency in LLM outputs, and provide valuable feedback for improving LLMs [19, 20] further.

---

[†]Corresponding Author.

38th Conference on Neural Information Processing Systems (NeurIPS 2024).

Verification models generally fall into two training paradigms: outcome supervision and process supervision. In outcome supervision, the training annotations rely on the correctness of the final answer [21, 22], while in process supervision, the annotations are based on evaluations of each reasoning step [23, 19]. However, process supervision is demanding in terms of annotations. Typically, it relies on either expensive and highly skilled human evaluators [23, 16] or model-induced process annotations [18, 17] to estimate the future correctness of the current reasoning step using Monte Carlo tree search [24, 25]. In contrast, outcome supervision only requires annotations for the output, making it more economical in terms of annotation effort but less effective. That being said when answers involve multiple reasoning paths, all aforementioned model-induced methods require numerous samples to ensure accurate estimations.

In this paper, we introduce **Auto**mated **P**rocess-**S**upervised **V**erifier (**AUTOPSV**), a novel approach that synergistically combines the strengths of both process supervision and output supervision. Our method begins by training an outcome-supervised verification model using outcome supervision annotations. This model then assigns confidence scores to each intermediate reasoning step, estimating their likelihood of contributing to a correct final answer. A distinguishing feature of AUTOPSV is its ability to automatically generate process annotations through relative step-level confidence change analysis, significantly reducing annotation effort while maintaining supervision quality without requiring ground truth answers. These automatically generated process annotations subsequently serve as training data for developing an enhanced verification model that leverages both process-level and outcome-level supervision signals. The complete framework of AUTOPSV is illustrated in Figure 1. We conduct extensive experiments across five datasets, including mathematical reasoning benchmarks and commonsense reasoning tasks. The results demonstrate that our method effectively improves the reasoning capability of the model with our highly efficient labeling scheme for process supervision. Our contribution is summarized as follows:

- We introduce AUTOPSV to automate the labeling of process data to enhance LLMs' reasoning capabilities. By combining the strengths of output and process supervision, AUTOPSV effectively identifies variations in model confidence to annotate the correctness of intermediate reasoning steps, enabling efficient automatic labeling for process supervision.

- Comprehensive experiments demonstrate that AUTOPSV significantly improves the performance and scalability of verification models in mathematical and commonsense reasoning tasks. This approach greatly reduces the need for manual intervention and extensive computational resources, making it a valuable tool for enhancing LLM capabilities.

- AUTOPSV's versatility is evident in its applicability to both labeled and unlabeled dataset settings after completing the training process. This flexibility and generalizability highlight the method's potential for widespread adoption in various LLM applications.

## 2 Related Works

**Improving Reasoning Abilities of LLMs**    To enhance the reasoning capabilities of LLMs, prior research primarily focuses on specific prompting techniques [26]. Existing efforts include few-shot prompting with intermediate steps augmented demonstrations [5–7, 27] or zero-shot prompting with specific instructions [28, 29]. Although these methods have shown promising results, their effectiveness is often constrained by their task-specific nature and the labour-intensive process of designing prompts, leading to inconsistent outcomes across different tasks [9, 10]. Another strategy to facilitate reasoning involves instruction tuning or knowledge distillation, which elicits reasoning paths from LLMs without explicit prompting [11–13, 30]. These approaches typically involve resource-intensive fine-tuning over LLMs and require a large set of examples annotated with chain-of-thoughts (CoT). Unlike methods that directly modify parameters or prompts, AUTOPSV focuses on training an additional verification model to select the desired output from the original model's output. This approach is further discussed in the context of process supervision in the following paragraph.

**From Outcome to Process Supervision**    Recent efforts have focused on enhancing the reasoning capabilities of LLMs through the use of verifiers to select the best answer from multiple candidates. There are two main types of verifiers: the Outcome-Supervised Verifier (OSV) and the Process-Supervised Verifier (PSV). The OSV is supervised with a signal based on the final answer [21, 22], while the PSV is with detailed feedback which requires evaluating individual reasoning steps [15,

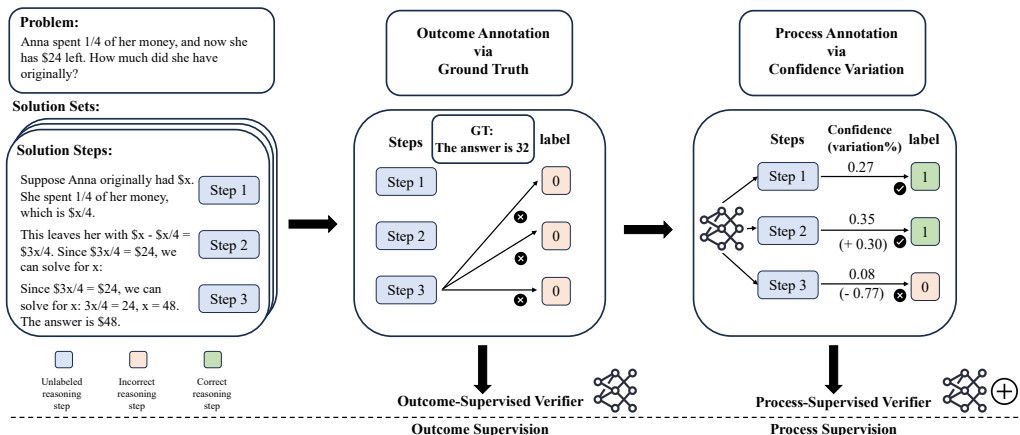

Figure 1: An overview of AUTOPSV. It utilizes an outcome-supervised verifier to automatically generate process annotations for each reasoning step by detecting its own confidence variations, *without* relying on ground truth annotations. AUTOPSV efficiently produces annotations serving as process supervision during the LLM training, which sidesteps costly annotations.

19, 16, 23]. Despite the time-consuming annotation cost, the PSV offers several advantages that make it preferable to the OSV. It can provide fine-grained feedback by pinpointing the location of errors, which is valuable for reinforcement learning and automatic correction [16, 20]. To alleviate the extensive human annotation, recent approaches [17, 18] propose a machine annotation framework using Monte Carlo Tree Search [24, 25]. This process demands a lot of computing resources, potentially imposing a limitation on the usage. AUTOPSV is more efficient, as it utilizes an outcome-supervised verification model to assign confidence to each reasoning step and calculate relative step-level confidence changes, eliminating the need for additional sampling or manual labeling.

## 3 AUTOPSV

In this section, we introduce the main problem that this paper focuses on in Section 3.1. We then discuss the motivation behind why we believe it is necessary to train a verification model in Section 3.2. Finally, we describe how we accomplish the transition from outcome supervision to process supervision during the training of the verification model in Section 3.3.

### 3.1 Problem Setting

**Objective** Our research addresses the challenge of response selection from multiple candidates generated by a Large Language Model (LLM). Specifically, given an LLM acting as a *response generator*, we seek to develop an effective method for identifying the correct response among multiple generated solutions. Our primary goal is to maximize the probability of selecting an accurate solution from the available candidates.

**Notation** We define $q$ as the input question presented to the model, and $S_i^{(1:t)}$ as the sequence of intermediate reasoning steps up to step $t$ for the $i$-th solution to question $q$. The binary correctness label for the $i$-th solution is denoted as $y_i$, and $\mathcal{Q}$ represents our training question dataset. This formalization enables us to systematically approach the response selection problem while maintaining mathematical rigor in our methodology.

### 3.2 Motivation

Response selection methods can be broadly divided into two categories: models specifically fine-tuned for the selection task and those that employ various prompting strategies.

In our exploration, we initially investigate whether existing open-source LLMs could serve as effective selection agents to evaluate model outputs and choose the correct response without fine-tuning. We

choose Mixtral-Instruct [31] as the *response generator*, and its response results are listed Table 1. Our objective focuses on identifying the correct response from **five** candidate solutions.

To establish robust conclusions, we evaluate *selector* models ranging from 7B to over 70B parameters, applying various prompting strategies. The results, tested on the GSM8K test set [21], are presented in Table 2. Notably, even models exceeding 70 billion parameters demonstrate suboptimal selection performance when relying solely on prompting without fine-tuning.

Table 1: Performance of Mixtral-Instruct on GSM8K. All results are reported in accuracy (%).

| Response Generator | Model Size (Parameters) | Pass@1 (%) | Pass@5 (%) | Self-Consistency (%) |
|---|---|---|---|---|
| Mixtral-Instruct [31] | 8 x 7B (MOE) | 62.55 | 82.31 | 69.06 |

Table 2: Comparison of different selection methods across various model sizes for selecting a response from candidate responses generated by Mixtral-Instruct. All results are reported in accuracy (%).

| Selector | Model Size | Prompt Strategy | | | | |
|---|---|---|---|---|---|---|
| | | Pairwise | Classification | Classification + CoT | Scoring | Scoring + CoT |
| Mistral-Instruct [32] | 7B | 60.73 | 61.18 | 64.82 | 61.49 | 69.75 |
| Mixtral-Instruct [31] | 8×7B | 58.83 | 59.14 | 67.40 | 61.79 | 65.58 |
| Llama2-chat [33] | 70B | 59.28 | 62.70 | 66.79 | 59.74 | 62.93 |
| Qwen [34] | 72B | 59.14 | 66.64 | 69.52 | 61.86 | 65.88 |

Based on these findings, our research focuses on training dedicated verification models and enhancing their response selection capabilities.

## 3.3 Training Methodology

AUTOPSV integrates outcome supervision with process supervision to create an effective training methodology. Below, we detail each component of our approach.

**Outcome-Supervision** We begin by training an outcome-supervised verification (OSV) model, denoted as $f_{\boldsymbol{\theta}}(\cdot)$, where $\boldsymbol{\theta}$ represents the optimized parameters. The training utilizes Mean Squared Error (MSE) loss for each solution step:

$$\mathcal{L}(S_i^{(1:t)}, y_i; q) = \left( f_{\boldsymbol{\theta}}(S_i^{(1:t)}; q) - y_i \right)^2 \tag{1}$$

The complete objective function across all training questions $\mathcal{Q}$ is:

$$\mathcal{L}_{\text{total}}(\mathcal{Q}) = \frac{1}{|\mathcal{Q}|} \sum_{q \in \mathcal{Q}} \frac{1}{n} \sum_{i=1}^{n} \sum_{t=1}^{m_i} \left( f_{\boldsymbol{\theta}}(S_i^{(1:t)}; q) - y_i \right)^2 \tag{2}$$

Where $n$ represents solutions per question and $m_i$ denotes steps in the $i$-th solution. The OSV output approximates the expected correctness probability, as formalized as follows:

**Theorem 1** *For a model trained with outcome supervision, $f_{\boldsymbol{\theta}}$, characterized by optimally tuned parameters $\boldsymbol{\theta}$, the assigned score for the sequence $S^{(1:t)}$ is an estimation of the likelihood of ultimately deriving a correct answer, denoted by $\hat{a}$, based on the progression observed in $S^{(1:t)}$ and the pertinent question $q$. This is mathematically represented as:*

$$f_{\boldsymbol{\theta}}(S^{(1:t)}; q) \approx p(\hat{a}|S^{(1:t)}, q)$$

The proof follows from optimizing the MSE loss in equation (2), with details in [22].

**Process-Supervision** We compute the relative confidence change between steps:

$$\Delta_{conf}^t = \frac{f_{\boldsymbol{\theta}}(S^{(1:t+1)}; q) - f_{\boldsymbol{\theta}}(S^{(1:t)}; q)}{f_{\boldsymbol{\theta}}(S^{(1:t)}; q)} \tag{3}$$

$\Delta_{conf}^t$ represents the relative variation in the model's confidence score from step $t$ to step $t+1$. A negative value of $\Delta_{conf}^t$ signifies a reduced confidence in achieving a correct answer after incorporating information from the $(t+1)$-th step. We denote the process label for the $t$-th step as $y_i^t$ and $\theta$ as the variation threshold.

For process labeling, we employ the "first error location" strategy [15] with threshold $\theta$:

- If $\Delta_{conf}^t > \theta$: $y_i^t = 1$
- Otherwise: $y_i^t = 0$ and $\forall t' > t, y_i^{t'} = 0$

The process-supervision loss function is:

$$\mathcal{L}_{proc}(S_i^{(1:t)}, y_i^t; q) = \left( f_{\boldsymbol{\theta}}(S_i^{(1:t)}; q) - y_i^t \right)^2 \tag{4}$$

## 4 Preliminary Findings

In this section, we present our findings aiming to validate two key aspects: In Section 4.1, we present a comprehensive analysis of the OSV model, i.e., to validate that the initially trained OSV model is effective and robust. In Section 4.2, we further introduce a self-designed benchmark for process errors and calculate $\Delta_{conf}^t$ to detect these errors, i.e., to demonstrate the effectiveness and reliability of relative step-level confidence change in the proposed method. The validation of these two components serves as a foundation for automatic process labeling via AUTOPSV, as described in Section 3.3.

### 4.1 Experiment on Outcome-Supervised Verifier Performance

In this section, we validate the effectiveness and scalability of the OSV model. Initially, we fine-tune a pretrained language model using ground truth data from the GSM8K dataset. Then, we use this fine-tuned model to generate multiple response samples for the GSM8K training prompts. We label these samples based on the correctness of their final answers. After this, we train an OSV model using the method described in Eq. (1).

To evaluate the OSV, we measure its ability to select a sample with the correct final answer from samples generated by various LLMs, denoted as the *Response Generator*. Specifically, our task involves selecting the correct candidate from **five** responses. We assess the effectiveness of outcome supervision on two models: Phi2 (**OSV (Phi)**) [35] and Mistral-7B (**OSV (Mistral)**) [32]. To explore the scalability of this outcome-supervised verifier effect, we choose Response Generators of varying scales, ranging from 7B to 72B parameters, i.e., Mistral-7B-Instruct (**Mistral-Instruct**) [32], Mixtral $8 \times 7B$ (**Mixtral-Instruct**) [31] and Qwen-72B-Chat (**Qwen**) [34]. This allows us to check the OSV's generalized ranking capability across different LLM scales.

Table 3: Performance of OSV models across different configurations.

| Response Generator | Pass@1 | Pass@5 | SC | OSV (Mistral) | OSV (Phi) |
|---|---|---|---|---|---|
| Mistral-Instruct | 42.08 | 69.90 | 50.03 | 60.72 | 52.61 |
| Mixtral-Instruct | 62.55 | 82.31 | 69.06 | 74.07 | 69.37 |
| Qwen | 77.03 | 91.13 | 81.27 | 85.00 | 84.19 |

The results demonstrate the effectiveness and scalability of the OSV model in selecting the correct response among multiple responses generated by different generators. Specifically, the OSV models, trained using either Mistral or Phi, consistently outperform the self-consistency (**SC**) baseline across all generator configurations. The results validate the effectiveness of the OSV model in enhancing model selection strategies, particularly when applied to larger and more accurate LLM generators.

We further analyze the performance discrepancy between the two OSV models:

**Performance Analysis of Different OSVs**   The performance disparity among the verifiers can be attributed primarily to variations in model sizes and the quality of their training data. It's important to note that the OSV model is **continuously** trained from the GSM8K fine-tuned model with the

addition of a value head. This means that the training data for each OSV model is generated from its corresponding fine-tuned base model. Specifically, the training data for OSV (**Mistral**) is generated from the fine-tuned **Mistral** model, while the training data for OSV (**Phi**) is generated from the fine-tuned **Phi** model.

Table 4: Model sizes and training data accuracy for training OSVs.

| Verifier | Size | Training Data | |
|---|---|---|---|
| | | Quality (acc.%) | Quantity (per question) |
| OSV (Mistral) | 7B | 0.9914 | 100 |
| OSV (Phi) | 2.7B | 0.9605 | 100 |

Table 4 presents a consolidated view of the model sizes along with the precision metrics of their outcome supervision training data.

It is worth noting that while both models are trained on the same quantity of data per question (100 samples), the quality of this data differs due to the capabilities of their respective base models. The Mistral model, being larger and potentially more capable, generates higher quality training data for its OSV, which in turn leads to better performance. In our content, we select the OSV (**Mistral**) model as the OSV model among other experiment settings due to its superior performance, as demonstrated Table 3.

## 4.2 Detecting Calculation Error During Math Reasoning

In this section, we verify the effectiveness and reliability of our method AUTOPSV. Specifically, We calculate $\Delta_{conf}^{t}$ to identify inaccuracies in the process, as outlined in Eq. (3).

In Section 4.2.1, we introduce the concept of math calculation error and establish a preliminary benchmark. In Section 4.2.2, we assess the performance of calculating $\Delta_{conf}^{t}$ to detect calculation errors against our established benchmark.

### 4.2.1 Math Calculation Error

We outline a method for identifying instances of math calculation error, which we define as occurrences where the numerical values on either side of an equals sign within a mathematical expression do not align. This misalignment indicates a breakdown in logical reasoning, categorizing the instance as a calculation error in the context of mathematical problem-solving. This process establishes a benchmark for math calculation error detection with more details in Appendix E.

**Math Calculation Error Detection** To identify calculation errors in mathematical reasoning, we monitor the relative step-level confidence changes between the intermediate steps as defined in Eq. (3). if $\Delta_{conf}^{t} \leq \theta$, we view the step as "incorrect". We also provide a detection example in Figure 2 for better understanding.

### 4.2.2 Quantitative Results

We introduce three metrics for a thorough evaluation of math calculation error detection: Precision (**Prec.**), which calculates the proportion of samples with correct final answers but exhibiting hallucinatory errors during the reasoning process; **Recall**, which determines the proportion of samples with math calculation errors that the OSV model successfully identifies through step-level confidence changes; **F1-score**, which gauges the verifier's overall efficacy. In Table 5, we explore how different threshold ($\theta$) values affect the precision, recall, and F1-score for math calculation error detection.

The results in Table 5 demonstrate that our method using step-level confidence change effectively detects calculation errors across threshold values from - 0.5 to - 0.9. As the threshold becomes more negative (stricter for labeling errors), the precision increases, indicating higher precision in identifying true errors. However, the recall decreases, meaning fewer actual errors are caught. Importantly, the F1-score, balancing precision and recall, remains relatively stable across thresholds. This demonstrates that our method strikes a good balance between detecting real errors and minimizing incorrect

Table 5: Process Calculation Error Detection Performance with Varying Threshold ($\theta$) Values.

| Metric | Threshold ($\theta$) Value | | | | |
|---|---|---|---|---|---|
| | 0.5 | 0.6 | 0.7 | 0.8 | 0.9 |
| **Prec.** | 0.85 | 0.88 | 0.91 | 0.93 | 0.94 |
| **Recall** | 0.90 | 0.89 | 0.86 | 0.83 | 0.80 |
| **F1-Score** | 0.88 | 0.89 | 0.88 | 0.88 | 0.86 |

flagging of valid calculations. Overall, our detection method is effective and robust, performing well over a range of thresholds without significantly compromising overall detection quality.

We note that setting $\theta$ = - 0.5 in our detection methods helps maintain a balance between precision and recall, which can ensure a balanced distribution of labeled "incorrect" and "correct" responses.

**Validation and Foundation for AUTOPSV** Our empirical validation of the OSV model encompasses two key aspects: its efficacy in response selection (Section 4.1) and its capability in detecting calculation errors (Section 4.2). These experimental results provide a robust foundation for automating process annotations using AUTOPSV. Furthermore, Theorem 1 establishes the theoretical framework for utilizing OSV to estimate the probability of reaching correct final answers from any given intermediate reasoning step. This convergence of theoretical guarantees and empirical evidence provides the methodological groundwork for applying AUTOPSV to generate process-supervised training data in our subsequent experiments.

## 5 Experiment

In this section, we first introduce the experimental setup in a subsection, which includes the response generator LLMs and evaluation settings in Section 5.1. We then present the main result of our process supervision-enhanced verification model on both mathematical and commonsense reasoning benchmarks, as described in Section 5.2.

### 5.1 Experimental Setup

**Models:** We selected three instruction-tuned LLMs of varying sizes, ranging from 7 billion to over 70 billion parameters, to serve as the *response generator*. Specifically, we used Mistral-Instruct-7B (**Mistral-Instruct**), Mixtral-8x7B-Instruct-v0.1 (**Mixtral-Instruct**), and Qwen-72B-Chat (**Qwen**).

**Datasets:** Our evaluation encompasses five benchmarks across two domains: For mathematical reasoning, we include GSM8K [21], containing math word problems requiring multi-step reasoning, and MATH [36], composed of high school-level competition problems covering a range of math subjects. For commonsense reasoning, we use HellaSwag [37], a dataset for physically situated commonsense reasoning, Winogrande [38], fill-in-the-blank problems requiring commonsense pronoun resolution and ANLI [39], a dataset for natural language understanding and reasoning.

**Evaluation:** For evaluation, we follow the methodology outlined in [40] to ensure consistency across benchmarks. Our protocol involves:

(i) Our object is to select the correct answer from five candidate responses.

(ii) For OSV models, we use the final value assigned to the whole solution for evaluation, while for PSV models, we utilize the product of step-level scores as the aggregation function.

(iii) To obtain more reliable pass@k results, we implement the estimation method described in [41]. This involves generating n samples per task (where n > k) and assessing the number of correct samples that pass unit tests. We then calculate the unbiased estimator for pass@k. For self-consistency (Self-Cons.) and verifier results, we randomly select k out of n samples and perform separate calculations. All results are reported with an accuracy of ±0.1 at a 95% confidence level.

Additional details regarding generation hyperparameters and different aggregation functions are provided in Appendix F.2.

## 5.2 Enhanced LLMs Reasoning via Process Supervision

To evaluate the efficacy and scalability of our proposed approach (detailed in Section 5.2), we conducted comprehensive experiments across mathematics and commonsense reasoning tasks using five diverse datasets.

Our experimental framework employs an autonomous process-supervision data annotation method where we calculate $\Delta_{conf}^{t}$ based on model confidence from OSV, using a threshold of $\theta$ = -0.5. This annotated data is then utilized to continuously fine-tune the OSV model, resulting in our enhanced OSV + PSV model.

We evaluate three distinct approaches: (i) Self-Consistency (**Self-Cons.**): Our baseline approach (ii) Outcome-supervised verifier (**OSV**): Our initial verification model (iii) Process-supervised enhanced verifier (**OSV + PSV**): Our proposed enhancement. For each approach, we assess Pass@5 performance, which represents the upper limit of achievable performance on these benchmarks.

**Mathematics Reasoning:** As shown in Table 6, the process-supervised enhanced verifier demonstrates superior performance over the outcome-supervised verifier and Self-Consistency models for all evaluated response generators on GSM8K. For the MATH benchmark, the process-supervised enhanced verifier outperforms the other two approaches for Mistral-Instruct and Mixtral-Instruct, but it is slightly less effective than the Self-Consistency model when applied to Qwen-72b.

Table 6: Results on mathematics benchmarks.

| Response Generator | GSM8K | | | | MATH | | | |
|---|---|---|---|---|---|---|---|---|
| | Pass@5 | Self-Cons. | OSV | OSV + PSV | Pass@5 | Self-Cons. | OSV | OSV + PSV |
| Mistral-Instruct | 69.90 | 50.03 | 61.18 | **61.41** | 7.7 | 1.64 | 5.10 | **5.30** |
| Mixtral-Instruct | 82.30 | 69.06 | 74.91 | **76.04** | 22.80 | 10.66 | 15.2 | **16.92** |
| Qwen | 91.13 | 81.27 | 84.91 | **85.15** | 56.10 | **40.10** | 38.94 | 39.36 |

**Commonsense Reasoning:** According to Table 7, OSV + PSV again leads to the best results among the three methods for each response generator tested on HellaSwag. For Winogrande, Mistral-Instruct paired with OSV + PSV achieves the highest performance, whereas, for Mixtral-Instruct and Qwen-72b, the original OSV without process supervision has a marginal advantage. When looking at the results of the ANLI benchmark, OSV + PSV is the highest-performing method for Mistral-Instruct and Mixtral-Instruct. Despite this, for Qwen-72b, the OSV model alone falls slightly behind the integrated OSV + PSV.

Table 7: Results on commonsense reasoning benchmarks.

| Response Generator | HellaSwag | | | | Winogrande | | | | ANLI | | | |
|---|---|---|---|---|---|---|---|---|---|---|---|---|
| | Pass@5 | Self-Cons. | OSV | OSV + PSV | Pass@5 | Self-Cons. | OSV | OSV + PSV | Pass@5 | Self-Cons. | OSV | OSV + PSV |
| Mistral-Instruct | 76.84 | 40.30 | 73.81 | **74.45** | 91.16 | 58.64 | 79.16 | **79.98** | 73.4 | 45.6 | 59.8 | **59.3** |
| Mixtral-Instruct | 84.05 | 73.67 | 82.83 | **83.62** | 79.16 | 68.75 | 73.40 | **73.88** | 68.4 | 59.0 | 62.9 | **64.0** |
| Qwen-72b | 95.28 | 85.44 | 93.08 | **93.99** | 88.63 | 72.21 | **80.34** | 79.32 | 82.4 | 63.8 | 69.1 | **71.4** |

**Conclusion:** Our experimental results indicate that the process-supervised enhanced verifier (**OSV + PSV**) consistently outperforms or matches the baseline models across most mathematical and commonsense reasoning tasks. By leveraging automatic process annotations, our approach enhances the model's capacity to verify reasoning processes, resulting in improved accuracy and robustness across a wide range of benchmarks and response generators.

## 6 Analysis

In Section 6.1, we compare our process annotation method, AUTOPSV with two other model-induced annotation methods to showcase the effectiveness and efficiency of our proposed approach. In Section 6.2, we validate the data quality constructed via AUTOPSV as described in Section 5.2.

## 6.1 Advantages of AutoPSV in Labeled and Unlabeled Settings

Aside from the labeling method defined by Eq. (3) in AUTOPSV, another labeling strategy is the Monte Carlo Tree sampling estimation (MCTS), as described in [17, 18]. To better demonstrate the effect of our method, we make a comparison with this approach and conduct experiments on mathematical benchmarks across both labeled and unlabeled settings (i.e., whether the ground-truth answers to the questions are provided).

**Performance in Labeled Settings**  We first evaluate the performance of AutoPSV in labeled settings. We follow the experimental settings described in [17, 18] to ensure a fair comparison. More implementation details are provided in Appendix F.3. Table 8 presents a comparison of process labeling methods across different response generators on the GSM8K and MATH datasets.

Table 8: Comparison of process labeling methods' performance across different response generators on GSM8K and MATH datasets. The table evaluates the Pass@5, Self-Consistency (Self-Cons.), and response selection performance of models fine-tuned using process annotations labeled by MCTS and AUTOPSV.

| Response Generator | GSM8K | | | | MATH | | | |
|---|---|---|---|---|---|---|---|---|
| | Pass@5 | Self-Cons. | Process (MCTS) | Process (AUTOPSV) | Pass@5 | Self-Cons. | Process (MCTS) | Process (AUTOPSV) |
| Mistral-Instruct | 69.90 | 50.03 | 54.13 | 55.32 | 7.7 | 1.64 | 3.3 | 3.24 |
| Mixtral-Instruct | 82.30 | 69.06 | 72.36 | 72.12 | 22.80 | 10.66 | 12.18 | 12.54 |
| Qwen-72b | 91.13 | 81.27 | 82.17 | 82.83 | 56.10 | 40.10 | 36.88 | 37.10 |

The experimental results shown in Table 8 suggest that our proposed method for process labeling, which relies on detecting changes in model confidence, performs competitively with the MCTS method from [18, 17]. In some cases, our method even outperforms the MCTS method, especially on the more challenging MATH benchmark.

A key advantage of AutoPSV is its computational efficiency. As shown in Table 9, our method requires significantly fewer tokens for process labeling compared to MCTS-based methods.

Table 9: Comparison of annotation costs between MCTS and AUTOPSV for process labeling on the GSM8K and MATH datasets. Annotation cost represents the processed tokens into a model when generating process annotations, encompassing both input and output tokens.

| Dataset | #Questions | #Solution Statistical | | | | Annotation Cost | |
|---|---|---|---|---|---|---|---|
| | | #Steps(Avg.) | #Steps(Overall) | #Tokens(Avg.) | #Tokens(Overall) | Process (MCTS) | Process (AUTOPSV) |
| GSM8K | 7,473 | 4.47 | 334,358 | 126 | 9,379,258 | 2,808 | 127 |
| MATH | 7,498 | 16.00 | 1,200,177 | 272 | 1,621,515,894 | 21,626 | 273 |

This efficiency stems from AutoPSV's ability to generate process annotations without requiring multiple samples for each reasoning step, making it particularly suitable for large-scale applications or scenarios with limited computational resources.

**Performance in Unlabeled Settings**  To further demonstrate the flexibility of AutoPSV, we evaluate its performance in unlabeled settings. We generated an additional dataset of 7,000 unlabeled math problems using the Evol-Instruct method from WizardLM [42]. These problems, created by LLMs without accompanying gold solutions, represent a challenging scenario for traditional supervision methods. We then conducted an experiment incorporating these unlabeled questions alongside the GSM8K dataset. Table 10 compares various methods across different response generators. In this context, OSV+PSV (GSM8K) refers to the original AutoPSV setting, while OSV+PSV (GSM8K+WizardLM) includes process annotations sourced from both GSM8K and WizardLM unlabeled questions. Notably, both MCTS and OSV-only training cannot leverage these unlabeled data, highlighting another key advantage of AutoPSV.

Table 10: Performance enhancement of our proposed AutoPSV method in unlabeled settings, where both MCTS and OSV-only training are unable to utilize unlabeled data.

| Response Generator | Pass@5 | Self-Cons. | OSV (GSM8K) | MCTS (GSM8K) | OSV+PSV (GSM8K) | OSV+PSV (GSM8K+WizardLM) |
|---|---|---|---|---|---|---|
| Mistral-Instruct | 69.90 | 50.03 | 61.18 | 60.82 | 61.41 | 63.11 |
| Mixtral-Instruct | 82.30 | 69.06 | 74.91 | 75.10 | 76.04 | 78.15 |
| Qwen | 91.13 | 81.27 | 84.91 | 84.85 | 85.15 | 86.77 |

The results demonstrate that the addition of unlabeled data leads to noticeable improvements across all response generators. For instance, the performance of Mistral-Instruct improves from 61.18 (OSV) to 63.11 (OSV+PSV with GSM8K+WizardLM). These results further underscore the value of the AutoPSV approach, particularly its ability to effectively utilize unlabeled data for enhanced performance.

In summary, AutoPSV offers several key advantages over MCTS-based methods:

**Consistent Performance with Computational Efficiency:** AutoPSV demonstrates robust, consistent improvements across different response generators and datasets. Its ability to efficiently generate process annotations—without the need for extensive sampling like MCTS—makes it particularly well-suited for large-scale applications and environments with limited computational resources.

**Leveraging Unlabeled Data for Enhanced Performance:** Unlike MCTS, which relies on ground truth labels, AutoPSV can effectively utilize unlabeled data. This capability not only enhances model performance in real-world settings but also offers scalability in scenarios where labeled data is scarce. The flexibility to harness unlabeled data ensures that AutoPSV can drive significant improvements even in data-constrained situations.

## 6.2 Outcome-Supervised Verification vs. Process-Supervised Verification

We apply the OSV to relabel the process-supervised training data as in Section 5.2. We then retrain a new model using this relabeled data. This experiment highlights the performance gap between outcome-supervised and process-supervised training.

Table 11: Experimental results showing the performance of OSV models across different configurations tested on GSM8K test sets.

| Response Generator | Pass@1 | Pass@5 | SC | OSV | PSV |
|---|---|---|---|---|---|
| Mistral-Instruct | 42.08 | 69.90 | 50.03 | 60.72 | 59.14 |
| Mixtral-Instruct | 62.55 | 82.31 | 69.06 | 74.07 | 71.39 |
| Qwen-72b | 77.03 | 91.13 | 81.27 | 85.00 | 83.70 |

The experimental results in Table 11 reveal that retraining the model with process supervision from AUTOPSV still yields better performance than self-consistency across three different response generators. We also noticed a small performance gap between the PSV and OSV. It is worth noting that our PSV was trained using data from the OSV. The small performance gap between the PSV and OSV models demonstrates that the relabeled process-supervised training method successfully inherits information from the outcome-supervised model without requiring ground truth annotations. This ablation study further provides quality assurance for automatic process labeling via AUTOPSV. Moreover, OSV training is limited to labeled datasets, while AUTOPSV demonstrates superior performance by utilizing both labeled and unlabeled data as shown in Table 10. This comparison further highlights the versatility and effectiveness of AUTOPSV in real-world scenarios where ground truth annotations may be scarce or unavailable.

## 7 Conclusion

In conclusion, we propose a novel method for automatic process labeling in LLMs by detecting relative changes in model confidence. Our experimental results demonstrate that AUTOPSV significantly enhances the precision and scalability of the verifier models in various reasoning tasks, ranging from mathematical to commonsense reasoning. AUTOPSV therefore has the potential to considerably enhance existing LLMs' performance while drastically reducing the need for intensive computation and manual intervention. For future work, we aim to utilize the automatically constructed PSV to supervise the generator using step-wise proximal policy optimization, to enhance the accuracy of the generator's output during greedy decoding without the need for subsequent reranking. This avenue of research could lead to even more advancements in the capabilities of LLMs and their application in reasoning tasks. The limitations and broader impact of the paper are discussed in Appendix A and B.

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

## A  Limitations

AUTOPSV is a promising solution for enhancing the reasoning capabilities of LLMs. However, it is important to acknowledge several potential limitations of the method. Firstly, while AUTOPSV aims to reduce the need for manual intervention, there is still a risk of inaccurate annotations. The relative step-level confidence change used to produce process annotations is an estimation and may not always accurately represent the actual correctness of a reasoning step. This could compromise the quality of the annotations and, in turn, the effectiveness of the method. Secondly, the success of AUTOPSV is heavily dependent on the performance of the verifier. If the model is not accurate enough in its step-level scores, the quality of the process annotations generated by AUTOPSV could be compromised. Thirdly, AUTOPSV is specifically designed to improve the reasoning capabilities of LLMs. Therefore, its applicability may be limited to tasks that involve complex multi-step reasoning. It is unclear how well the method will scale or generalize to other tasks or domains that do not involve intensive reasoning. This is an important consideration for future research and development of the method.

## B Broader Impact

**Positive Societal Impacts**   The proposed AUTOPSV has the potential to bring about several positive societal impacts. By enhancing the reasoning capabilities of LLMs, AUTOPSV can lead to more accurate and reliable information, which in turn can support better decision-making in various sectors, including healthcare, education, and finance. Moreover, AUTOPSV combines the strengths of output supervision and process supervision to automatically annotate reasoning steps, significantly reducing the time, effort, and cost associated with manual annotation. This makes the process of training LLMs more efficient and accessible. Additionally, the process supervision data generated by AUTOPSV can improve the performance and scalability of verification models, allowing for the development of more complex and sophisticated LLMs capable of handling a wider range of tasks and applications.

**Positive Societal Impacts**   However, AUTOPSV also presents several potential negative societal impacts. The automation of the annotation process could lead to job displacement for individuals currently employed in this role. There is also a risk that AUTOPSV and the enhanced LLMs could be misused, for instance, to spread misinformation or manipulate public opinion. The increased reliance on LLMs for decision-making could potentially result in a decrease in critical thinking and problem-solving skills among individuals. Furthermore, the use of LLMs in various sectors could lead to privacy and security issues, as these models often require large amounts of data for training.

## C More Related Work

**Learning From Feedback**   Improving LLMs through learning from feedback has become a prevalent strategy, notably through reinforcement learning from human feedback, which seeks to align LLMs with human values by refining their outputs based on feedback [43–45]. However, this method faces challenges such as high costs due to manual labor and a lack of real-time feedback capabilities [46, 47]. An alternative strategy involves using self-correcting LLMs, which rely on automated feedback to iteratively adapt and understand the consequences of their actions without heavy reliance on human intervention. This feedback can be derived from outside sources such as other models [48, 49], tools [50, 51], knowledge bases [52, 53], evaluation metrics [54, 55] or generation logits [56].

External feedback leverages external perspectives to identify errors and verify factual accuracy, offering insights that may not be recognized by the LLM alone. Conversely, feedback can also be internally generated, where the LLM evaluates and refines its output iteratively until a desired quality is achieved [57–60]. This self-improvement mechanism is particularly valuable in scenarios where external feedback is scarce or restricted [61, 62]. However, recent effort [63] suggests that LLMs struggle to independently identify and correct errors through self-generated prompts.

## D Vanilla Evaluation Methods Description

**Classification:** For this method, the evaluator is presented with multiple answers for a given question and is required to choose the best answer among them. The selection is made based on the evaluator's judgment of which answer most accurately addresses the question or provides the most relevant information.

> Given the following question: '[question]', and these five answers:
> 1. [answer]
> 2. [answer]
> 3. [answer]
> 4. [answer]
> 5. [answer]
> Which answer is the best? Please provide the number of the best answer. You should strictly follow the output format requirements and not output any other content.
> Example: Answer [number] is better. Let's Begin!

**Classification + COT:** In Classification + COT, the evaluator must not only identify the best answer but also analyze and compare all provided answers before making their decision. This method

requires a deeper examination of the content and context of each answer to determine its quality and relevance to the question.

> Given the following question: '[question]', and these five answers:
> 1. [answer]
> 2. [answer]
> 3. [answer]
> 4. [answer]
> 5. [answer]
> Which answer is the best? Please analyze and compare the provided answers and then identify the number of the best answer. You should strictly follow the output format requirements and not output any other content.
> Example: Comparison and Analysis: [analysis]. Best answer: [number]. Let's Begin!

**Scoring:** In the Scoring method, the evaluator assigns a numerical score to each answer based on its quality or relevance to the given question. The scores typically range from 1 to 10, with 10 representing the highest quality or most relevant answer.

> Given the following question: '[question]', please score these five answers on a scale from 1 to 10, where 10 is the best:
> 1. [answer]
> 2. [answer]
> 3. [answer]
> 4. [answer]
> 5. [answer]
> Please provide a score for each answer. You should strictly follow the output format requirements and not output any other content.
> Example: Answer i: [score]. Let's Begin!

**Scoring + COT:** Similar to Scoring, Scoring + COT also involves assigning numerical scores to each answer. However, in Scoring_cot, the evaluator is required to provide an analysis of each answer before assigning a score. This analysis informs the scoring process and ensures a more informed evaluation of each answer.

> Given the following question: '[question]', please score these five answers on a scale from 1 to 10, where 10 is the best:
> 1. [answer]
> 2. [answer]
> 3. [answer]
> 4. [answer]
> 5. [answer]
> Please analyze each answer, and then provide a score for each answer. You should strictly follow the output format requirements and not output any other content.
> Example: Answer i: analysis: [analysis]. score: [score]. Let's Begin!

**Pairwise Comparison:** In this method, the evaluator is presented with pairs of answers for a given question and is tasked with determining which answer is better in each pair. The evaluator compares the content or quality of each answer and selects the one they deem superior. The Pairwise Comparison method differs from other evaluation methods in that it evaluates two candidate answers at a time and chooses the winner to proceed to the next comparison with the next candidate answer. For a set of n candidates, this method conducts n-1 pairwise comparisons. To mitigate the potential order preference bias exhibited by LLMs, we adopt a method similar to [64] which shuffles the positions of two answers during prompting. This ensures a fair evaluation process by eliminating any bias toward the position of the answers.

> Given the following question: '[question]', compare each pair of answers and decide which one is better:
> Compare 1. [answer1] with 2. [answer2]
> For the comparison, indicate the better answer with its number. You should strictly follow the output format requirements and not output any other content.
> Example: Answer i is better. Let's Begin!

By employing these different evaluation methods, we aim to comprehensively assess the quality and relevance of the answers generated by our models for various questions. Each method offers a unique perspective and contributes to a more thorough evaluation process.

# E   Math Calculation Error Benchmark

**Methodology**   We utilize the LlaMA2-chat model (LlaMA) for mathematical reasoning step generation. Using regular expressions, we extract computational steps and evaluate expressions to the left of the "=" sign using Python's *eval* function to verify their correctness against right-hand side results. We term this process "Math Calculation Error Detection" and present the results in Table 12.

Table 12: Performance of the LlaMA model on GSM8K training data, evaluated through few-shot cot prompting strategy.

| Model | Pass@5 (%) | Self-Consistency (%) | Math Calculation Error Detection (%) |
|-------|------------|----------------------|--------------------------------------|
| LlaMA | 0.4791     | 0.2881               | 0.1824                               |

**Data Processing**   Non-computational expressions (e.g., "x + 1 + 2 = 4") that cannot be evaluated due to unsolvability or incorrect formatting are excluded from the ground truth data. This ensures our ground truth accurately reflects only computational errors during reasoning. While DeepMind [15] employs human labelers for "trace errors" annotation, our automated approach using Python's *eval* function provides a scalable alternative.

**Error Detection Example**   To demonstrate our calculation of $\Delta^t_{conf}$ for math calculation error identification, we present an example in Figure 2. The red highlights indicate calculation errors in the reasoning process, detected through significant decreases in model confidence.

# F   Experiment Details

## F.1   Training Hyperparameters

**Computing Infrastructure**   Our experiments were conducted using 8 NVIDIA A100 GPUs, each with 40GB of memory. All models underwent full-parameter fine-tuning using the AdamW optimizer.

**Verifier Training Configuration**   For both process-supervised and outcome-supervised methods, we maintain consistent training parameters as detailed in Table 13. The training process spans 1 epoch with a batch size of 512 and a learning rate of $2 \times 10^{-6}$, incorporating a 3% learning rate warmup period.

Table 13: Verifier Training Hyperparameters.

| Hyperparameter | Global Batch Size | LR | Epo. | Max Length | Weight Decay | Warmup Ratio |
|----------------|-------------------|----|------|------------|--------------|--------------|
| **Value**      | 512               | $2 \times 10^{-6}$ | 1 | 2048 | 0 | 0.03 |

**SFT Model Prerequisites**   Prior to verifier training, we establish a supervised fine-tuning (SFT) model. This model serves to generate responses and facilitate outcome supervision labeling by validating final answer correctness. The SFT model's training parameters are specified in Table 14. For comprehensive details regarding OSV implementation, refer to [22].

Figure 2: **A data example during math calculation error detection.** We apply the OSV model to detect calculation errors by calculating the step-level confidence change at each step, denoted inside the brackets [ ] for the step containing calculations. The ground truth location of the math calculation error is marked in red.

Table 14: SFT Training Hyperparameters.

| Hyperparameter | Global Batch Size | LR | Epo. | Max Length | Weight Decay | Warmup Ratio |
|---|---|---|---|---|---|---|
| **Value** | 128 | $5 \times 10^{-6}$ | 2 | 2048 | 0 | 0.03 |

## F.2 Generation Settings

**Parameter Settings** We employ two distinct temperature configurations: temperature = 0.0 for the greedy decoding strategy and temperature = 0.7 for the pass@k evaluation strategy.

**Pass@k Evaluation Protocol** To ensure unbiased estimation of pass@k metrics, we follow the methodology established in [41]. We generate $n = 20$ samples per problem instance and evaluate each sample against unit tests to determine correctness. The unbiased estimator for pass@k is then computed based on these results. This entire process is repeated 5 times to establish 95% confidence intervals, as detailed in Section 5.1.

**Score Aggregation Methodology** Our primary aggregation mechanism employs a multiplicative approach for step-level scores. This method computes the product of confidence scores across all steps, yielding a compound probability that represents the overall likelihood of solution correctness.

**Comparative Analysis of Aggregation Functions** Following [16], we conducted a comparative analysis of different aggregation strategies. While our primary approach utilizes the product of step-level scores, we also investigate the minimum of step-level scores as an alternative approach. The comparative results for these aggregation functions on the GSM8K dataset are presented below:

As shown in Table 15, the choice of aggregation function can have a slight impact on the results, with the product function generally performing comparably or slightly better than the minimum function across different response generators.

Table 15: Comparison of aggregation functions on GSM8K dataset.

| Response Generator | Product | Minimum |
|---|---|---|
| Mistral-Instruct | 60.72 | 61.17 |
| Mixtral-Instruct | 74.07 | 72.45 |
| Qwen | 85.00 | 84.28 |

### F.3 Process-Supervision Labelling Strategy

**Overview**    Our implementation encompasses two distinct labelling methodologies: our proposed approach (**Process Ours**) and a comparative Monte Carlo Tree Search approach (**Process MCTS**).

**Implementation Procedure**    The procedure consists of the following sequential stages:

1. **Initial Generator Training:** We conduct supervised fine-tuning of a generator using a combined dataset of approximately 15,000 problems from GSM8K and MATH.

2. **Sample Generation:** The trained generator produces 10 distinct solution attempts for each unlabeled prompt within the training datasets. Each solution receives a binary label (0 or 1) based on the correctness of its final answer.

3. **Method-Specific Processing:**

   (a) **Process Ours:**
      - Train an output-supervised verifier using the complete set of 150,000 generated samples (15,000 × 10)
      - Relabel samples using a relative confidence change criterion (as defined in Equation 3)
      - Retrain the verifier using these refined labels

   (b) **Process MCTS:**
      - Decompose each of the 150,000 samples into constituent reasoning paths
      - For each incomplete reasoning path, generate eight complete path variations
      - Compute the accuracy for each complete path
      - Calculate the proportion of correct paths to determine the final label
      - Retrain the verifier using these MCTS-derived labels

**Technical Note**    While our main experimental results employ 50 samples per problem, we restrict this implementation to 10 samples due to computational constraints. The full MCTS labeling process for 50 × 15,000 samples would require approximately 10-12 days of computation time.

