# OpenReview forum: "AutoPSV: Automated Process-Supervised Verifier"
_NeurIPS.cc/2024/Conference — NeurIPS 2024 poster_

### Official Review · Reviewer_hKWR · 2024-07-05

**Soundness:** 2
**Presentation:** 2
**Contribution:** 2
**Rating:** 7
**Confidence:** 4

**Summary:**

This paper introduces AutoCV, a novel method to create process reward models (PRMs). The approach involves training an an outcome-supervised verification model based on (S_{(1:t)}, y) pairs where S_{(1:t)} is the 1st~t-th steps in the response and y is the final correctness label of the entire response. The outcome-supervised verification model then generates process annotations by detecting confidence variations across reasoning steps. AutoCV combines outcome and process supervision, reducing the need for manual annotations or computationally expensive methods. Experiments across five datasets in mathematics and commonsense reasoning show a significant advantage of the PRM for best-of-N sampling. The method outperforms self-consistency baselines and outcome-supervised verifiers while requiring fewer tokens for annotations than existing approaches.

**Strengths:**

- This work proposes a novel method to obtain process annotation for free from outcome supervision. The method is elegant and derived from a theoretical perspective.
- Unlike previous work on PRM that only focuses on the math domain, this work shows effectiveness on commonsense reasoning. and natural language inference.

**Weaknesses:**

Though the method is interesting, the experiments are not very convincing and the paper is not well presented.
- I cannot get the point of section 4.2.1 since it's a bit confusing to me. Is there a reason why the method is specifically useful for "process calculation hallucination detection"? It seems to me that it's a general method to detect all faulty steps in a response, which is what a PRM is supposed to do. Also, I think the term "hallucination" may be overloaded, which originates from an open-ended generation. I suppose the setting in section 4.2.1 is simply "calculation error detection".
- There is no baseline in the main experiment (Table 6&7). I'd suggest moving section 6.1 to the main exp, and results beyond the math domain as a separate subsection in section 5.
- The setup part (sec 5.1) does not mention if the experiments are all in-domain. OOD evaluations are required to examine the generalizability.
- From Table 8, AutoCV performs very comparable with MCTS-based methods with differences of <1% accuracy. The performance gain is marginal.
- Table 9 may not be a fair comparison of efficiency between AutoCV and MCTS-based methods. Despite using fewer tokens, AutoCV requires training an ORM/OSV for annotation which could also be costly. Maybe the sum of GPU hours for training/inference would be fairer?

**Questions:**

- It seems the verifiers are based on the Mistral series and in general its gain is less significant when the response generator is Qwen, especially on MATH where AutoCV best-of-5 underperforms self-consistency. Is it because the method lacks generalizability or does the method only work well when the response generator is weak?

**Limitations:**

See above

---

> ### Author Rebuttal · Authors · 2024-08-06
>
> Thank you for your thoughtful review and insightful comments. We hereby address your concerns below:
>
>
>
> **R W1 Presentation of Section 4.2.1:**
>
> Thanks for the suggestion. To avoid potential misunderstandings, we will change the term to "calculation error detection."
>
> We would like to provide the following clarification for Section 4.2.1. We introduce a method to **automate** the acquisition of ground truth by detecting errors in mathematical calculations. This approach provides an effective and robust automated framework for detecting math calculation errors **without requiring manual annotations**. Based on this framework, we further establish a PCH benchmark as described in Appendix E. This addresses the primary challenge of evaluating whether a model detects all faulty steps due to the lack of ground truth process annotations.
>
>
>
> **R W2 No baseline in the main experiment (Table 6 & 7)**
>
> We will move the baselines from Section 6.1 to the main experiments (Tables 6 & 7) and separate the math and common reasoning tasks into different subsections for better presentation.
>
>
>
> **R W3 OOD evaluations are required to examine the generalizability**
>
> To address your concern regarding the need for out-of-domain (OOD) evaluations, we utilized the trained OSV+PSV model from the original manuscript. We conducted additional OOD tests on MMLU, BBH, and ARC-C. As shown in the table below, our method consistently outperforms baselines on three benchmarks.
>
> |                        | MMLU   |           |      |         | BBH    |           |      |         | Arc-C  |           |      |         |
> | ---------------------- | ------ | --------- | ---- | ------- | ------ | --------- | ---- | ------- | ------ | --------- | ---- | ------- |
> | **Response Generator** | Pass@5 | Self-Cons | OSV  | OSV+PSV | Pass@5 | Self-Cons | OSV  | OSV+PSV | Pass@5 | Self-Cons | OSV  | OSV+PSV |
> | Mistral-Instruct       | 62.5   | 47.4      | 52.5 | 54.3    | 40.5   | 30.7      | 35.5 | 36.1    | 52.5   | 42.3      | 48.3 | 49.1    |
> | Mixtral-Instruct       | 69.6   | 58.1      | 63.4 | 64.7    | 43.3   | 38.4      | 40.2 | 40.8    | 58.1   | 50.1      | 53.9 | 54.3    |
> | Qwen-72b               | 76.1   | 67.7      | 71.8 | 71.9    | 65.9   | 58.3      | 60.9 | 61.3    | 64.7   | 56.3      | 60.2 | 61.5    |
>
>
>
> **R W4/5 Comparisons with MCTS-based Methods:**
>
> 1. GPU Hours Comparison: we provide a comparison of GPU usage for both training and annotation costs:
>
> | Dataset | GPU Hours (Annotation Cost) |                  | GPU Hours (Training) |                  |
> | ------- | --------------------------- | ---------------- | -------------------- | ---------------- |
> |         | Process (MCTS)              | Process (AutoCV) | Process (MCTS)       | Process (AutoCV) |
> | GSM8K   | 64                          | 3                | 2.5                  | 5                |
> | MATH    | 480                         | 6                | 2.5                  | 5                |
>
> As the number of steps required to solve a question increases (e.g., MATH compared to GSM8K), the time required for MCTS annotation increases **quadratically**. Considering training and annotation costs, the total GPU hours for AutoCV are approximately **1/8** of those for MCTS on GSM8K and approximately **1/40** for MATH. We will update Table 9 to reflect this and provide a more balanced view of the efficiency of AutoCV compared to MCTS-based methods.
>
>
>
>
>
> 2. Flexibility and Further Performance Enhancement: MCTS-based methods **require ground truth** on the correctness of final answers. In contrast, AutoCV generates process annotations by detecting relative confidence variations without needing ground truth annotations. This flexibility allows AutoCV to utilize redundant unlabeled questions, improving the model's performance. The results are provided below:
>
> | **Response Generator** | Pass@5 | Self-Cons. | OSV (GSM8K) | MCTS(GSM8K) | OSV+PSV (GSM8K) | OSV+PSV (GSM8K+WizardLM) |
> | ---------------------- | ------ | ---------- | ----------- | ----------- | --------------- | ------------------------ |
> | Mistral-Instruct       | 69.90  | 50.03      | 61.18       | 60.82       | 61.41           | 63.11                    |
> | Mixtral-Instruct       | 82.30  | 69.06      | 74.91       | 75.10       | 76.04           | 78.15                    |
> | Qwen                   | 91.13  | 81.27      | 84.91       | 84.         | 85.15           | 86.77                    |
>
> When MCTS and OSV-only training are limited to GSM8K labeled data, our AutoCV extends to apply 7K unlabeled math problems following the Evol-Instruct method from WizardLM [1]. Including unlabeled data further enhances the model's capabilities, demonstrating the value of the AutoCV approach.
>
>
>
> **R Q1 Verifier Performance with Different Generators**
>
> We applied the Llama2-13B [2] model and followed the settings in our paper, testing the MATH test set. The results are presented below:
>
> | **Response Generator** | Pass@5 | Self-Cons. | OSV+PSV (Mistral-7B) | OSV+PSV (LLaMA-13B) |
> | ---------------------- | ------ | ---------- | -------------------- | ------------------- |
> | Mistral-Instruct       | 7.70   | 1.64       | 5.30                 | 6.13                |
> | Mixtral-Instruct       | 22.80  | 10.66      | 16.92                | 20.19               |
> | Qwen                   | 56.10  | 40.10      | 39.36                | 43.13               |
>
> From the table, it is evident that increasing the model size to 13B improves the performance of AutoCV across all settings. This demonstrates that our method works effectively even with strong response generators like Qwen, not just weaker ones. It shows that AutoCV performs better when applied to stronger and larger LLMs.
>
>
>
> [1] [WizardLM: Empowering Large Language Models to Follow Complex Instructions. ICLR 2024](https://arxiv.org/abs/2304.12244)
>
> [2] [Llama 2: Open Foundation and Fine-Tuned Chat Models. Arxiv 2023](https://arxiv.org/abs/2307.09288)

---

> > ### Comment · Reviewer_hKWR · 2024-08-08
> > **Reviewer Response**
> >
> > Thanks for your clarification.
> >
> > Most of my concerns have been addressed, but I think AutoCV still needs ground truth for verifier training. Yes, the process annotation stage does not require so, but the number of annotated data for verifier training may have an effect on the final result.
> >
> > Anyway, I am pretty sold by the approach now and will raise my score. I would love to further increase the score if the authors can show the effects of verifier capability on the final process reward annotation.

---

> > > ### Author Response · Authors · 2024-08-11
> > > **Impact of Annotated Data on Verifier Capability and Final Performance**
> > >
> > > Thank you for your feedback. We have conducted additional experiments to investigate how the quantity of annotated data affects the verifier's capability and its impact on the final outcomes.
> > >
> > > 1. **Verifier Capability:** We trained the OSV on subsets of the annotated data, specifically using 25%, 50%, and 75% of the full dataset. We then evaluated its performance in detecting mathematical calculation errors, applying the same metrics defined in Section 4.2.2 with a threshold of $\theta = -0.5$. The results obtained using the full training dataset are provided below, as also presented in Table 5 of the original manuscript.
> > >
> > > **OSV Performance in Detecting Mathematical Calculation Errors:**
> > >
> > > |              | **25%** | **50%** | **75%** | Full |
> > > | ------------ | ------- | ------- | ------- | ---- |
> > > | **Accuracy** | 0.78    | 0.82    | 0.84    | 0.85 |
> > > | **Recall**   | 0.81    | 0.86    | 0.88    | 0.90 |
> > > | **F1-Score** | 0.79    | 0.83    | 0.87    | 0.88 |
> > >
> > > **Experiment 1 Analysis:** When the training data exceeds 1/2 of the full dataset, the incremental gains become less pronounced, with the performance at 3/4 of the data closely approaching that of the full dataset. This suggests that our method remains effective even with a moderate amount of annotated data, making it adaptable to scenarios where the availability of labeled data is limited.
> > >
> > > 2. **Impact on Final Outcomes:** Additionally, we evaluated the final performance of our OSV + PSV combination on both the GSM8K and MATH test sets.
> > >
> > > **Final Performance of OSV + PSV on the GSM8K Test Set:**
> > >
> > > |                        | **25%** |           | **50%** |           | **75%** |           | **Full** |           |
> > > | ---------------------- | ------- | --------- | ------- | --------- | ------- | --------- | -------- | --------- |
> > > | **Response Generator** | OSV     | OSV + PSV | OSV     | OSV + PSV | OSV     | OSV + PSV | OSV      | OSV + PSV |
> > > | Mistral-Instruct       | 58.12   | 60.13     | 59.66   | 60.72     | 60.45   | 61.10     | 61.18    | 61.41     |
> > > | Mixtral-Instruct       | 71.14   | 73.75     | 72.53   | 74.39     | 73.93   | 75.52     | 74.91    | 76.04     |
> > > | Qwen                   | 80.59   | 82.93     | 82.47   | 84.10     | 84.01   | 84.83     | 84.91    | 85.15     |
> > >
> > > **Final Performance of OSV + PSV on the Math Test Set:**
> > >
> > > |                        | **25%** |           | **50%** |           | **75%** |           | **Full** |           |
> > > | ---------------------- | ------- | --------- | ------- | --------- | ------- | --------- | -------- | --------- |
> > > | **Response Generator** | OSV     | OSV + PSV | OSV     | OSV + PSV | OSV     | OSV + PSV | OSV      | OSV + PSV |
> > > | Mistral-Instruct       | 4.13    | 4.41      | 4.55    | 4.72      | 4.95    | 5.15      | 5.10     | 5.30      |
> > > | Mixtral-Instruct       | 12.59   | 13.87     | 13.97   | 14.50     | 14.80   | 16.52     | 15.20    | 16.92     |
> > > | Qwen                   | 35.20   | 36.91     | 37.10   | 38.54     | 38.13   | 39.01     | 38.94    | 39.36     |
> > >
> > > **Experiment 2 Analysis:** A similar trend is observed in Experiment 2, where the impact of annotated training data becomes even less pronounced when combining OSV and PSV. As the verifier's capability improves with more training data, the incremental gains in the final performance of the OSV + PSV combination decrease after the training data surpasses 50% of the full dataset. For example, on the GSM8K test set, the difference in performance between using 75% and the full dataset with OSV alone is 0.73 (60.45 vs 61.18), whereas, with the OSV + PSV combination, this difference reduces to just 0.31 (61.10 vs 61.41). Similarly, this effect is consistently observed on the MATH test set. This indicates that our approach remains not only effective with a moderate amount of ground-truth-labeled data but also that **the combined OSV + PSV system is particularly robust, showing even smaller performance drops when the training data is reduced.** This further reinforces the adaptability and efficiency of our method in scenarios where labeled data for the OSV are limited.
> > >
> > >
> > >
> > > **Overall Conclusion:**   These results confirm that while more annotated data enhances verifier performance, our approach remains robust and effective even with limited data. The combined OSV + PSV system shows minimal performance degradation with reduced training data and **can also utilize unlabeled questions to improve its performance**, making it well-suited for practical applications where labeled data is limited.
> > >
> > > We will incorporate this into the revised findings to provide a comprehensive evaluation of our method. We would love to provide any further details if you have more questions. Thank you again for your thoughtful review.

---

### Official Review · Reviewer_rE8z · 2024-07-11

**Soundness:** 3
**Presentation:** 3
**Contribution:** 2
**Rating:** 6
**Confidence:** 4

**Summary:**

This paper proposes a new method (AutoCV) that bridges the gap in popular techniques for enhancing reasoning capabilities of LLMs. Prior work have proposed *verification models* (models that evaluate generated reasoning steps and rerank candidate responses) as a promising solution. Here the community has focused on two training paradigms for verification models: (1) outcome supervision (train on correctness of final answer; cheap but less effective) and (2) process supervision (train on labels for each reasoning step; expensive but effective). AutoCV attempts to bridge these two by first training an outcome-supervised model and then use it to annotate confidence for unannotated intermediate steps. Then, by calculating the relative confidence variation between steps, AutoCV generates process supervision data that can be used for training.

**Strengths:**

AutoCV tackles clear limitations in prior approaches with some novel ideas:
1. Interesting use of how an outcome-supervised (OS) model (theoretically) implicitly learns process annotations:
     - OS replicates final answer’s correctness label across all steps, causing the model to implicitly learn values for partial paths.
     - Consequently, forces the optimal solution under MSE to be estimate of the probability of reaching the correct answer.

2. Overcomes clear efficiency limitations of previous automated process annotation approaches like MathShepherd and MiPS that use MCTS. The MCTS soln is to sample multiple complete traces at each reasoning step, and using the outcome to score the potential of this step (say k/N lead to correct answer).

**Weaknesses:**

My concerns are mainly around the presentation, clarity, and significance of some results:

**Presentation/Clarity**
1. Throughout the paper, there are several experiments using verifier at inference to rerank generations from a *response generator*. For instance, Section 4.1 and Table 3. However, there is no mention or analysis of the #samples being reranked?

2. Similarly, there is no mention (or analysis) of the aggregation function used for process-supervised models at inference time?

3. Section 4.1 and Table 4 attribute the performance disparity among OSVs (for Phi2 and Mistral) to training data quality. Particularly confusing, was why the training data was different at all for each model? From the description, it seems like the training data was generated from _a single_ (GSM8K fine-tuned) pretrained LLM. If not, how does using a stronger model to generate data affect performance?

**Significance of results.** Table 6-7 describe the performance of a PSV finetuned on self-generated confidence variations of an OSV. From that, it seems like there is *very minimal gains* from using the PS data generated to enhance (finetune) the OSV model. Furthermore, when the PS data is used to retrain a new PSV, the performance even drops a few points (Table 10).

1. While one can argue that there is "minimal loss" in information from the OSV to a PSV, it concerns me that the quality of process labels might not be good. The generated labels add little-to-no value to overall performance? i.e., why use AutoCV to train a PSV at all if OSV is equally effective AND can provide process-annotations scores?

2. With minimal gains, the only advantage of AutoCV is that it is more sample efficient than MCTS based approaches for PSV training. Are there contributions beyond efficiency?

**Questions:**

Note: Added most questions in the Weaknesses section. Additional:

Could the authors comment on how this would work with Agent based reasoning traces? Particularly, the procedure to generate process supervision labels (described in Section 3.3) would not work because it makes the following assumption: `any step, including the final result, following a step containing an error is considered incorrect`. This does not hold true for agentic approaches that use environment feedback to correct their mistakes (think code generation w/ execution feedback).

**Limitations:**

Yes

---

> ### Author Rebuttal · Authors · 2024-08-06
>
> Thank you for your thoughtful review and insightful comments. We hereby address your concerns below:
>
> **Response to W1 Presentation:**
>
> **1.1 No mention of the #samples being reranked**
>
> Our task involves selecting the correct candidate from **five** responses. Therefore, we compare our method to the Pass@5 performance in each table, including Table 3 in Section 4.1. Specifically, we noted on line 256 that "Pass@5 represents the upper limit of performance."
>
> We will add this detail to the table captions and the main context to enhance clarity.
>
> **1.2 Analysis of the aggregation function**
>
> We utilize the **product of step-level scores as the aggregation function**. Additionally, we have analyzed different aggregation functions, specifically the **minimum** and the **product** of step-level scores, following the setting of [1]. Below, we present the results on GSM8K:
>
> | **Response Generator** | **Product** | **Minimum** |
> | ---------------------- | ----------- | ----------- |
> | Mistral-Instruct       | 60.72       | 61.17       |
> | Mixtral-Instruct       | 74.07       | 72.45       |
> | Qwen                   | 85.00       | 84.28       |
>
> The product aggregation function multiplies the confidence scores of each step, leading to a compounded probability that represents the overall likelihood of a correct response.
>
>
>
> **1.3 Why the training data was different in Table 4**
>
> The OSV model is **continuously** trained from the GSM8K fine-tuned model with the addition of a value head.
>
> Therefore, in Table 4, the training data for OSV (**Mistral**) is generated from the fine-tuned **Mistral** model, while the training data for OSV (**Phi**) is generated from the fine-tuned **Phi** model. This distinction in training data sources explains the performance disparity observed between OSV models for different response generators.
>
> To avoid any confusion, we will add further clarification in the manuscript.
>
>
>
> **Response to W2 Significance of results:**
>
> **2.1 Quality of process labels**
>
> Table 10 shows that retraining the model with process labels only from AutoCV yields better performance than self-consistency, indicating that these labels are of good quality and successfully inherit information from the outcome-supervised model without requiring ground truth annotations.
>
> Additionally, Table 5 in Section 4 (Preliminary Findings) further demonstrates the effectiveness and reliability of AutoCV's relative confidence variation in detecting errors during math reasoning.
>
>
>
>  **2.2 Why train a PSV if using PS data drops performance (Table 10) and OSV is equally effective and provides process-annotation scores?**
>
> The training data for the PSV model **does not utilize ground truth labels**; it uses process supervision data generated by detecting confidence changes between steps (as in Eq. 3). Therefore, in the OSV+PSV training phase, we can include both the GSM8K/MATH training set and unlabeled math problems.
>
> In Table 10, the PSV model **maintains performance close to the OSV model. Notice that OSV relies on ground truth labels** to determine the correctness of each solution, but PSV does not. To further illustrate the no-ground-truth benefits, we generated 7K unlabeled math problems following the Evol-Instruct method from WizardLM [1], which are generated by LLMs without gold solutions.  We conducted an additional experiment on GSM8K by including these unlabeled questions. **Methods like MCTS and OSV-only training cannot effectively utilize these unlabeled questions.**
>
> | **Response Generator** | Pass@5 | Self-Cons. | OSV (GSM8K) | MCTS(GSM8K) | OSV+PSV (GSM8K) | OSV+PSV (GSM8K+WizardLM) |
> | ---------------------- | ------ | ---------- | ----------- | ----------- | --------------- | ------------------------ |
> | Mistral-Instruct       | 69.90  | 50.03      | 61.18       | 60.82       | 61.41           | 63.11                    |
> | Mixtral-Instruct       | 82.30  | 69.06      | 74.91       | 75.10       | 76.04           | 78.15                    |
> | Qwen                   | 91.13  | 81.27      | 84.91       | 84.         | 85.15           | 86.77                    |
>
> MCTS and OSV-only training is limited to GSM8K training data, our AutoCV extends to apply unlabeled data, contributing to a performance gain (e,g, 61.41 to 63.11 for Mistral-Instruct). Including unlabeled data further enhances the model's capabilities, demonstrating the value of the AutoCV approach.
>
>
>
> **2.3 Advantage of AutoCV compared with MCTS-based methods beyond efficiency.**
>
> The advantage of AutoCV is not just efficiency over MCTS-based approaches but also the ability to leverage **unlabeled** data for enhanced performance. MCTS-based methods **require ground truth on the correctness of final answers**. In contrast, the process annotations in AutoCV are generated by detecting relative confidence variations without needing ground truth annotations. This flexibility allows AutoCV to utilize unlabeled data, improving the model's performance. The experiment result is provided in the response to **2.2.**
>
> **R Q1 How AutoCV work with Agent-based reasoning ?**
>
> Integrating AutoCV into agent-based reasoning traces is an interesting direction to explore. We can dynamically update labels based on feedback at each step instead of marking all subsequent steps as incorrect after detecting an error:
>
> 1. Detect the confidence variation $\Delta_{conf}^{t}$ as defined in Eq (3). If $\Delta_{conf}^{t}$ is below an error threshold, mark the step as incorrect.
> 2. Continue labeling subsequent steps as incorrect until $\Delta_{conf}^{t}$ exceeds a correct threshold.
>
> The correct and error threshold settings may depend on experimental findings, similar to how we choose the threshold in AutoCV, as shown in Table 5.
>
>
> [1] [WizardLM: Empowering Large Language Models to Follow Complex Instructions. ICLR 2024](https://arxiv.org/abs/2304.12244)
>
> [2] [Let's Verify Step by Step. ICLR 2024](https://arxiv.org/abs/2305.20050)

---

### Official Review · Reviewer_kK32 · 2024-07-18

**Soundness:** 3
**Presentation:** 3
**Contribution:** 3
**Rating:** 7
**Confidence:** 3

**Summary:**

The authors propose AutoCV, a method for solving multi-step reasoning tasks with chain-of-thought prompting that involving automatically labeling each step of the multi-step process based on its likelihood of leading to the correct outcome, and combining an outcome supervised classifier (OSV) and process supervised classified (PSV) to come up with a more effective method. Experiments on 2 math and 3 reasoning datasets show that OSV + PSV outperforms self-consistency as well as just OSV across the board, albeit by a small margin in several cases.

**Strengths:**

The paper is carefully written and provides good background for readers (like myself) not very familiar with the approaches considered. The idea of using the difference in the confidence of an OSV model at step t and at t+1 is interesting; it's perhaps a bit surprising that it actually works well, as it's not often easy to learn how good an early intermediate step is towards achieving the final goal.

The findings in section 4 seem to lay out a good justification for the design choices behind AutoCV, although I found the discussion in 4.2 (hallucination in math reasoning) harder to follow.

The performance on 5 benchmark datasets, as noted above, is a strength of the paper, even though in many cases the absolute improvement is small.

**Weaknesses:**

Not having familiarity with the area, I wasn't able to judge the novelty of the work and the substance (i.e., whether there is enough new material to warrant publication). It seems to me that f_theta() being a good indicator of the probability of reaching the correct final outcome is something known from reference [22]. The new part here is looking at the delta of f_theta() between time steps t and t+1, as done in Eq (3).

In terms of naming, I found the use of the phrase "confidence variation" not intuitive for referring to the change in the confidence from t steps to t+1. To me, confidence variation suggests how the confidence changes as one varies some parameter or some sampling, not as one takes another step. In a similar vein, I found "AutoCV" to not be an informative name for the system -- it doesn't connect well with the task of multi-step reasoning or with chain-of-thought prompting. That said, this is a subjective choice and it's up to the authors to decide if they want to look for a more intuitive name.

As noted above, I found the discussion in section 4.2 somewhat confusing.

Lastly, as also noted above, while OSV + PSV generally shows consistent gains over the OSV, the gains are often very small. This somewhat lowers the value of the proposed technique, but consistent gains remain a net positive.

**Questions:**

Please see weaknesses above.

**Limitations:**

Yes

---

> ### Author Rebuttal · Authors · 2024-08-06
>
> Thank you for your thoughtful review and insightful comments. We hereby address your concerns below:
>
> **R W1  Novelty and Substance of Our Paper**
>
> 1. **Theoretical Contribution:** While reference [22] demonstrates that $f_\theta$ can indicate the probability of reaching the correct final outcome, our contribution extends this by exploring the change in $f_\theta$, denoted as $\Delta_{conf}^{t}$, between time steps $t$ and $t+1$. Unlike [22], which applies $f_\theta$ for reranking during decoding, our AutoCV demonstrates the effectiveness and robustness of $\Delta_{conf}^{t}$ for error detection, as detailed in Section 4.
> 2. **Experimental Contribution:** We demonstrate the performance gains of applying $\Delta_{conf}^{t}$ for process-supervision training in Sections 5 and 6. AutoCV combines the strengths of output supervision and process supervision to automatically annotate reasoning steps, thereby enhancing model performance.
> 3. **Scalability:** Unlike [22] and other MCTS-based methods, which limit their experiments to math tasks, our method is generally applicable across both math and commonsense reasoning. Furthermore, unlike methods limited to settings where ground truth is available, AutoCV can leverage both labeled and unlabeled data, significantly expanding the dataset for training. This flexibility enhances the robustness and applicability of our approach. More details on this can be found in **R W4.**
>
>
>
> **R W2 Naming Issue**
>
> Thank you for the suggestion. To better reflect the concept of changes in confidence from step t to step t+1, we propose using the term **"Step-level Confidence Change**. This name more accurately conveys the idea of confidence change between consecutive steps.
>
> Similarly, to reflect these aspects more clearly, we propose renaming the system to **"AutoPSR: Automated Process-Supervised Reasoner"**. This name highlights the focus on automated process labeling within the context of multi-step reasoning and chain-of-thought prompting. We use AutoCV during the rebuttal for consistency, but we will **revise the term in the revised manuscript**.
>
> **R W3 Discussion in Section 4.2:**
>
> To address the confusion regarding Section 4.2, we provide further clarification.
>
> Section 4.2 aims to **automate** the evaluation of the effectiveness and robustness of detecting relative variations defined in AutoCV for generating process labels.
>
> In Section 4.2.1, we automate the acquisition of ground truth by detecting errors specifically in mathematical calculations. **This approach eliminates the need for manual process data ground truth.** We successfully labeled mathematical calculation errors in the outputs of Llama2 for 7500 GSM8K training data samples. This process, as stated in line 195, "establishes a benchmark for PCH detection." **To avoid potential misunderstandings, we will revise the term "hallucination in math reasoning" to "math calculation error" in the revised manuscript.**
>
> In Section 4.2.2, we show that our method performs well in detecting calculation errors, as evidenced by precision, accuracy, and F1 scores. As noted in line 209, "Table 5 demonstrates that our method using confidence variation effectively detects calculation errors." This validates our method as a reliable source of process supervision information, **establishing an experimental basis for automating process annotations in Section 5.**
>
> We will include the clarification above in the revised manuscript for better presentation.
>
>
>
> **R  W4 Small Gains with OSV + PSV compaed with OSV only:**
>
> It is important to highlight that these gains, while seemingly small, **are statistically significant**. We conducted statistical significance tests over five repetitions. Experimental results consistently produced p-values well below the significance level of 1e-6. For example, with Mixtral-Instruct, the t-statistic was -15.06 with a p-value of 1.09e-07 This statistical significance underscores that the improvements observed are not due to chance and validate the effectiveness of the proposed OSV + PSV approach.
>
> Moreover, it is essential to consider the broader applicability and additional benefits that OSV + PSV provides. **Unlike OSV-only, which is limited to settings with available ground truth, OSV + PSV can leverage both labeled and unlabeled data.** This capability allows for the utilization of a more extensive and diverse dataset, which is particularly advantageous in real-world scenarios where labeled data may be scarce.
>
> To illustrate the benefits, we generated 7K math problems following the Evol-Instruct method from WizardLM[1], **which are generated by LLMs without gold solutions**. Both MCTS and OSV-only training **cannot** leverage these unlabeled data. The results are as follows:
>
> | **Response Generator** | Pass@5 | Self-Cons. | OSV (GSM8K) | MCTS(GSM8K) | OSV+PSV (GSM8K) | OSV+PSV (GSM8K+WizardLM) |
> | ---------------------- | ------ | ---------- | ----------- | ----------- | --------------- | ------------------------ |
> | Mistral-Instruct       | 69.90  | 50.03      | 61.18       | 60.82       | 61.41           | 63.11                    |
> | Mixtral-Instruct       | 82.30  | 69.06      | 74.91       | 75.10       | 76.04           | 78.15                    |
> | Qwen                   | 91.13  | 81.27      | 84.91       | 84.         | 85.15           | 86.77                    |
>
> In this context, OSV+PSV (GSM8K) refers to the original AutoCV setting, while OSV+PSV (GSM8K+WizardLM) includes process annotations sourced from both GSM8K and WizardLM unlabeled questions. The addition of unlabeled data leads to noticeable improvements across all response generators. For instance, the performance of Mistral-Instruct improves from 61.18 (OSV) to 63.11 (OSV+PSV with GSM8K+WizardLM). These results further demonstrate the value of the AutoCV approach.
>
> [1] [WizardLM: Empowering Large Language Models to Follow Complex Instructions. ICLR 2024](https://arxiv.org/abs/2304.12244)

---

> > ### Comment · Reviewer_kK32 · 2024-08-13
> > **Re: Rebuttal by Authors**
> >
> > Thank you for the detailed and informative response!  I am happy to see your willingness to reconsider the system name and for your efforts in explaining section 4.2 better.  Also happy to hear that the small-looking gains are statistically significant; please make sure to include that in case it's not already mentioned.
> >
> > I remain in favor of accepting this paper.

---

### Decision · Program_Chairs · 2024-09-25

**Decision:**

Accept (poster)

**Comment:**

The submission has several strenghs for supporting its acceptance. First, the proposed AutoCV approach combines the strengths of output supervision and process supervision to automatically annotate reasoning steps, enhancing the performance in drawing correct outcomes, even though in some test cases the absolute improvement is small. Second, different from previous work on process reward models that only focuses on the math domain, AutoCV also shows effectiveness on commonsense reasoning and natural language inference. Finally, the  presentation of the submission is fine. Notable issues on the presentation have been addressed in rebuttal.